# MultiVENT:
# Multilingual Videos of Events
# with Aligned Natural Text

**Kate Sanders\***     **David Etter\***     **Reno Kriz\***     **Benjamin Van Durme**
Johns Hopkins University
Human Language Technology Center of Excellence
{ksande25, detter2, rkriz1, vandurme}@jhu.edu

## Abstract

Everyday news coverage has shifted from traditional broadcasts towards a wide range of presentation formats such as first-hand, unedited video footage. Datasets that reflect the diverse array of multimodal, multilingual news sources available online could be used to teach models to benefit from this shift, but existing news video datasets focus on traditional news broadcasts produced for English-speaking audiences. We address this limitation by constructing MultiVENT, a dataset of multilingual, event-centric videos grounded in text documents across five target languages. MultiVENT includes both news broadcast videos and non-professional event footage, which we use to analyze the state of online news videos and how they can be leveraged to build robust, factually accurate models. Finally, we provide a model for complex, multilingual video retrieval to serve as a baseline for information retrieval using MultiVENT.

## 1 Introduction

Information dissemination for current events has traditionally consisted of professionally collected and produced materials, leading to large collections of well-written news articles and high-quality videos. As a result, such materials form the basis for significant prior work in content analysis and retrieval [54, 20, 2, 15, 50]. Meanwhile, a high volume of event-centric content today is generated by non-professionals, such as on-the-scene witnesses to events who hastily capture videos and upload them to the internet without further editing. We propose that this contemporary landscape of news content can be leveraged by models to produce a more comprehensive understanding of events. News agencies have adapted to this shift, often collecting and incorporating this online content into official broadcasts, but news video datasets still do not typically address this new domain of event coverage.

In addition to focusing on traditional news sources, existing news video datasets predominantly consider content produced in English. This is consistent with common practices in video dataset collection: Collected videos and captions are recorded in English, and when multilinguality is considered, it is achieved by directly translating captions and transcripts [48, 22, 40, 19]. Because this data is originally produced for English speaking audiences, these multilingual datasets can contain unwanted content biases like "translationese" [6, 21]. As event-centric video content produced in other languages makes up a large portion of news videos online, we argue that including organic, multilingual content is necessary for a diverse and perspective-agnostic sampling of event coverage.

With these ideas in mind, we present MultiVENT, a dataset of **Multi**lingual **V**ideos of **E**vents with aligned **N**atural **T**ext that contains 2,396 diverse, event-centric videos and text descriptions

---

\*Equal contribution.

37th Conference on Neural Information Processing Systems (NeurIPS 2023) Track on Datasets and Benchmarks.

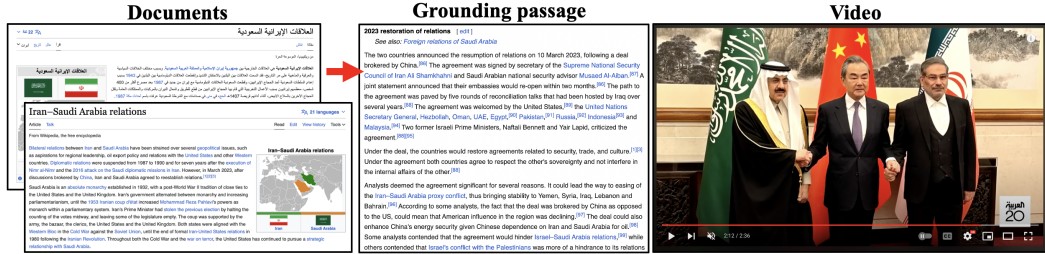

Figure 1: A sample video-text pair from MultiVENT. Every event-centric video is paired with a corresponding video description and a long-form text document describing the event, both in the same language as the video. If the language is not English, the video is also paired with a corresponding English document.

that reflect the distribution of news content online. The videos are grounded in natural language video descriptions and long-form text documents, and the data spans 260 current events across over forty countries. The content in MultiVENT is collected in five target languages: Arabic, Chinese, English, Korean, and Russian, and as the multilinguality is organic, the data is less likely to suffer from translation bias. We provide an illustration of the dataset's contents in Figure 1: Each natural language query (describing a video of a current event) is paired with grounding text documents and a unique corresponding video. We use MultiVENT to explore and characterize the variety of event-centric videos available online and illustrate the importance of leveraging these different video types when building multimodal information systems.

Citizen journalism, the most notable example being Wikipedia [18], has emerged alongside other online news sources as a method for curating comprehensive summaries of events. Work in natural language processing has considered the problem of automating this process by training models to generate informative reports using online source materials [24, 45, 33]. We use MultiVENT to explore how this process can be extended to incorporate multimodal sources of evidence. As a first step in this direction, we consider the task of video retrieval on MultiVENT, through which a model learns to retrieve multimodal source material given a natural language event description. This task differs from prior video retrieval benchmarks [9, 53, 1, 29, 51] as the videos in MultiVENT vary widely in length and content presentation, are multilingual, and can involve significant amounts of on-screen text. In addition to multilingual natural language captions for each video, we provide full text documents that ground the events and serve as more complex retrieval queries.

In summary, we address the lack of multilingual news video datasets that reflect the reality of our modern information landscape by proposing a dataset of diverse web videos depicting current events across five languages. Enumerated, our contributions are:

1. We present MultiVENT, a multimodal, multilingual information retrieval dataset of grounded videos depicting current events. The dataset targets five languages and covers a range of online video formats beyond traditional news broadcasts.

2. Using MultiVENT, we quantitatively illustrate the information presented by news videos and the differences in content between video formats, and qualitatively evaluate how multimodal coverage of an event can evolve over time.

3. We present MultiCLIP, a model for multilingual, event-centric video retrieval that serves as a baseline for video retrieval approaches on the task.

## 2 Related Work

### 2.1 Video retrieval datasets

Early video datasets generally contained short clips spanning narrow ranges of topics, such as the Microsoft Research Video Description Corpus [9]. Video datasets spanning larger domains include MSR-VTT [53] and DiDeMo [1], although the lengths of these videos were still relatively short. The V3C dataset [39, 3] offered longer video lengths while still spanning a wide range of topics such as news reports. A shift towards massive video datasets was instigated by HowTo100M [29], which included over 130 million video clips belonging to one million narrated instructional videos. VaTeX [48], released in the same year, considered video retrieval from a multilingual context using caption translation. Additional multilingual video retrieval datasets include Rudder [13], consisting of instructional videos for making toys with multilingual captions, MTVR [22], which extended the TVR dataset [23] by adding Chinese subtitles and queries, and Multi-HowTo100M [19], which extended HowTo100M by scraping YouTube for subtitles in up to 9 other languages. Recently, Chen et al. [8] released the ChinaOpen dataset which contains a wide range of video-caption pairs originally produced in Chinese. Recent work has also considered the problem of interpreting text-heavy video content: Wu et al. [51] and Jahagirdar et al. [20] introduced datasets that focus on within-video text and OCR annotations, including news broadcasts.

### 2.2 Video retrieval methods

The size of early video datasets allowed retrieval systems to rely on pre-extracted features from expert systems like action recognition models. As massive video datasets gained prominence, the video retrieval paradigm moved towards ad-hoc video-text feature extraction using large pretrained models. Dosovitskiy et al. [14] proposed using stand-alone transformer architectures for video understanding, and Bertasius et al. [7] showed that applying space- and time-based self-attention independently improved performance. Bain et al. applied findings directly to video retrieval, training and evaluating transformer architectures on WebVid-2M [4]. Radford et al. [34] introduced CLIP and showed that pretraining models to match captions to images can result in scalable models, and CLIP's applicability to video retrieval was demonstrated by Fang et al. [16] through their CLIP2Video model. More fine-grained modifications to CLIP were proposed. Wang et al. [47] introduced "Object-aware Transformers", which extended video-text transformers to incorporate object-level annotations within video footage, and Ge et al. [17] modified the pretraining task to involve teaching a vision-text model to answer multiple choice questions about a video. Bain et al. [5] adapted large image-text models to the task of long video retrieval by incorporating the weighted-mean of frame embeddings, and Wu et al. [51] incorporated independent optical character recognition and embeddings into the encoder pipeline to explicitly model in-video text.

### 2.3 Report generation using online sources

A wide range of research has used online corpora for report generation tasks, including QA-pair generation [36, 30] (such as SQuAD [37] and HotPotQA [55]) and knowledge graph generation [31]. Notably, Lewis et al. [24] introduced a method for automatically extracting question-answer pairs from large corpora of text documents, and applied this method to Wikipedia to produce the PAQ dataset. Some PAQ extensions have been multilingual — Pisare et al. [32] built the WikiOmnia QA dataset on Russian Wikipedia documents, and Rybak et al. [42] produced a question-Wikipedia passage dataset in Polish. Recently, Qian et al. [33] extended the ideas in PAQ to construct WebBrain, a task in which a model must generate factual articles with references given a natural language query. In the multimodal domain, Reddy et al. and Chen et al. have considered the problem of open-domain QA for image-text data [38, 10], with Chen et al. using Wikipedia to generate a multimodal dataset. In a similar vein, Li et al. propose a dataset for information extraction from multimedia articles [27] and an extraction approach that can be used with text, image, and video content [26].

## 3 Dataset

In this section we outline the MultiVENT collection process. We draw from Davidsonian event semantics [28] to define an event as a related set of particulars existing at a shared point in space

and time. We add the additional condition that an event must also be "newsworthy" in that it must have been reported on by respected news organizations. The dataset includes 2,396 videos and corresponding text descriptions covering 260 events grounded in 468 text documents, and includes content in Arabic, Chinese, English, Korean, and Russian. The average length per video is over one minute (83.7 seconds) and the full dataset contains over 55 hours of video content. We first identify 260 visually salient current events spanning from 2013 to 2023, and assign a target language to each event. Then, for each event, we collect grounding text documents and a set of videos in the event's target language.

## 3.1 Current event curation

We consider four primary event categories for MultiVENT: Disasters, political events, social events, and technology events. We include thirteen current events per category for each target language. We use Google Trends statistics to select these events, based on its tracking of term popularity based on internet activity by country. We construct lists of the top five countries with the most speakers of each target language and review the top trending topics on Google in each of these countries over the last ten years. We record topics and search phrases that corresponded to current events that (1) align with one of the predefined event categories and (2) have sufficient online video coverage. For categories that did not amass a sufficient list of current events per language through this process, we consult Wikipedia's yearly summaries of events to fill the remaining slots. Detailed statistics characterizing this set of current events are shown in Figure 2. As shown, the majority of selected events take place in the last few years, with only three taking place before 2016.

Also shown in Figure 2, there is not a bijective mapping between the language used in event coverage and the country the event took place in. The language and country are often related, e.g., Russian news content in MultiVENT predominantly takes place in Russia, but this is not true of all events in the dataset. For example, we include data in Chinese pertaining to the 2023 ATP tennis circuit in Dallas, Texas: At this event, tennis player Wu Yibing became the highest-ranked Chinese player in the history of the ATP rankings, and so the event received substantial Chinese news coverage. In cases such as this, news in multiple languages will heavily focus on the same current event, such as sports events and international political relations. We do not include the same event in multiple languages in MultiVENT by design, in contrast with data collection procedures used for efforts such as AIDA [46] which aim to cover a small collection of current events in many languages.

Every current event in the dataset is grounded in an English natural language document and, if the event is tagged with a non-English language, an additional natural language document in that target language. First, we check if a full English Wikipedia article exists for the current event. If not, we manually find a Wikipedia article that includes a passage describing the event. If Wikipedia does not have a passage that appropriately grounds the event, then a news article in English is selected as a grounding document instead. This process is then repeated for the target language. The dataset includes 468 grounding articles in total: 313 are full Wikipedia articles, 104 are Wikipedia passages, and 51 are external articles.

## 3.2 Video collection

We aim to collect visually and semantically distinct videos for each current event with an even split between firsthand witness accounts (e.g., first-person smartphone videos), amateur edited videos (e.g., vlogs), and professional news reports and compilations. Information regarding the resultant distribution of these categories and their semantic differences is included in Section 4.2. For each current event, we collect ten videos in the current event's target language. We search YouTube and Twitter for these videos using target keywords collected from the Google Trends search and Wikipedia. We check the upload date to ensure that it aligns with the time of the event's occurrence. Finally, we skim through the video content to confirm that it is relevant to the target event. After collecting the videos, we manually identify and remove duplicates, resulting in 2,396 videos in total. We do not include repeat videos, but sometimes professional news reports include firsthand footage that is already included as unedited footage in the dataset. In these cases, we keep both the news report and the original footage as the context and text metadata between the two are distinct. If the video has a natural language description, we tag the video with this description. If it does not, we use the video title as the tagged natural language description. We report the distribution of videos by source in Figure 2.

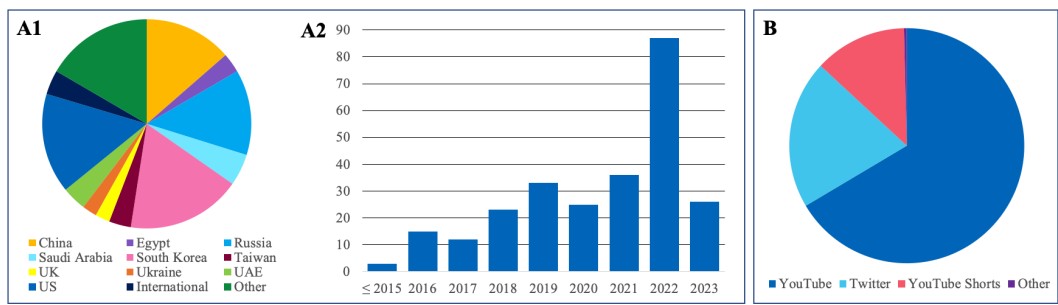

Figure 2: **A**: Statistics illustrating the distribution of current events selected for the dataset. (**A1**) depicts the general breakdown of countries in which each current event takes place. Many countries had a single current event, particularly small countries in the middle east and southeast Asia, and are consolidated into the "other" category for easier graph interpretation. (**A2**) shows the distribution of years during which the current events take place. **B**: Breakdown of data sources for the videos in the dataset. The majority of videos came from YouTube as it has a larger international audience and has existed longer than YouTube Shorts.

## 4 Data analysis

We present an analysis of MultiVENT to help characterize how online multimodal content contributes to our understanding of current events. We explore multimodal event coverage from three angles: (1) what kinds of information news videos contribute, (2) the differences in informative content provided by different types of news videos, and (3) how multimodal coverage of an event can evolve over time.

### 4.1 Semantic information in video

Visual data can provide rich, semantically nuanced details of an event that are not captured in text documents due to reporting bias, limitations of text, and document length limits. To characterize the complexity of these videos and the information they provide, we annotate a set of two hundred videos of disasters in MultiVENT to identify visual entities in the videos that help answer common "who, what, where"-type questions about the events they depict.

We present videos of disaster footage to local annotators and provide them with a set of event-centric questions derived from FrameNet's "disaster scenario" template [41]. We modify this template, designed for text, to sufficiently cover visual content (further information is included in the appendix). We instruct annotators to identify every on-screen entity (such as people, scrolling news headline banners, etc.) that might help answer one of these event-centric questions. These annotators are professionals with experience in data analysis and are compensated appropriately for their work. The annotations are for analysis only and are not part of the main dataset, and no personal information was captured.

The template divides salient entities into six categories: The disaster itself ("what"), the location of the disaster ("where"), the time the disaster takes place ("when"), people affected by the disaster ("who") and first responders for the disaster, e.g., firefighters (also "who"), and any visible outcomes of the disaster. Not every category applies to both visual content and text: We exclude "where" and "when" from the set of categories that visual content should be annotated for (because identifiable depictions of "where" are present in almost every frame, and "when" in virtually none) and disaster outcomes from the set of text annotation categories, as textual examples of this category tend to involve full clauses, which complicate the annotation process. We note that, while we exclude visual "where" content, different video types are likely to contain different amounts of visual information informing a viewer of where the event takes place. We omit this analysis for simplicity but do not model this facet of the videos' semantic richness by doing so.

We present the number of event-relevant entities that appear on-screen in these annotated videos in Table 1. For each annotated entity, we additionally ask annotators to rate their certainty that the entity is directly related to the event described by the video's natural language description from 0%

Table 1: Mean number of visual entities and in-scene text references (written text displayed within a video) present per video in a subset of 210 disaster videos from the current events dataset. We omit "where" and "when" entities from the visual content counts as "where" visual content technically appears in every frame and there are few types of visual evidence for "when" questions. We omit "outcomes" from the text references as an outcome by itself is a full event that is difficult to localize in text (this field is omitted from the FrameNet event template analogue for text documents).

| | Visual entities | Text references | Total |
|---|---|---|---|
| Disaster ("What") | 1.25 | 1.37 | **2.62** |
| Place of occurrence ("Where") | - | 1.54 | **1.54** |
| Time of occurrence ("When") | - | 0.77 | **0.77** |
| Affected people ("Who") | 1.22 | 0.54 | **1.76** |
| First responders ("Who") | 1.13 | 0.50 | **1.63** |
| Disaster outcomes | 1.00 | - | **1.00** |
| Total | **4.60** | **4.72** | **9.32** |

Table 2: Mean annotator certainty scores partitioned on entity type based on the annotations used for Table 1. 0.20 certainty indicates that the annotator is 20% sure that the annotated entity helps answer the tagged question about the described event, while 1.00 certainty indicates that the annotator is completely sure that the entity helps answer the tagged question about the event.

| | Disaster | Where | When | AP | FR | Outcomes | All |
|---|---|---|---|---|---|---|---|
| Visual content | .787 | - | - | .716 | .765 | .798 | .830 |
| Text content | .931 | .907 | .929 | .856 | .836 | - | .900 |
| Average | .862 | .907 | .929 | .759 | .787 | .798 | .865 |

to 100%. We record these certainty scores in 20% intervals, i.e. as 20%, 40%, 60%, 80%, or 100%. The averages of the linguists' confidence rankings by entity type are listed in Table 2.

As shown in Table 1, each video contains an average of 9.32 informative visual entities that pertain to the event in question. About half of these entities are purely visual, and half are within-video text that can be identified with an optical character recognition model. As indicated by Table 2, purely visual entities are more ambiguous than the text content shown onscreen alongside it, which aligns with past research that explores the difficulty humans have in interpreting visual content depicting complex events [43].

## 4.2   Video content by domain

As described in Section 3, we collect three main types of videos: Official news broadcasts, edited video footage, and raw, unedited footage. Of the 210 videos in the annotation set reported in Table 1, 53% are news broadcasts, 11% are edited footage, and 36% are raw footage. To quantify the difference in information presented by these different video types, we take the video annotations shown in Table 1 and partition these annotations by video type. We present the results in Table 3.

As shown by the results, news broadcasts depict the most relevant semantic information, followed by edited footage. This is particularly apparent when considering text content alone. On average, news coverage contains almost 9 times as much relevant on-screen text content than raw footage, and over three times more than edited footage. Visual content differences were less drastic, but news content still had two times more visual content than raw footage and 1.3 times more than edited footage. The difference in visual content between news coverage and edited footage is possibly due to average video length and the quality of the video curation — oftentimes, unprofessionally edited footage only draws from one source whereas news coverage draws from many.

## 4.3   Information evolution

As shown in Table 3, first-person footage is often opaque compared to professional coverage. However, comprehensive coverage often builds on earlier, less informative coverage. This can be seen in news

Table 3: Mean number of visual entities and in-scene text references present per video, partitioned on video type. Same 210 video subset is used for analysis as that used for the analysis shown in Table 1.

| | News coverage | | | Edited footage | | | Raw footage | | |
|---|---|---|---|---|---|---|---|---|---|
| | Vis. | Text | Total | Vis. | Text | Total | Vis. | Text | Total |
| Disaster ("What") | 1.42 | 2.38 | **3.80** | 1.14 | 0.41 | **1.55** | 1.05 | 0.17 | **1.22** |
| Place ("Where") | - | 2.51 | **2.51** | - | 0.59 | **0.59** | - | 0.39 | **0.39** |
| Time ("When") | - | 1.26 | **1.26** | - | 0.41 | **0.41** | - | 0.16 | **0.16** |
| Affected people ("Who") | 1.48 | 0.94 | **2.42** | 1.18 | 0.23 | **1.41** | 0.86 | 0.04 | **0.90** |
| First responders ("Who") | 1.73 | 0.78 | **2.51** | 1.36 | 0.45 | **1.81** | 0.17 | 0.12 | **0.29** |
| Disaster outcomes | 1.28 | - | **1.28** | 0.77 | 0.27 | **1.04** | 0.67 | - | **0.67** |
| Total | **5.91** | **7.87** | **13.78** | **4.45** | **2.36** | **6.81** | **2.75** | **0.88** | **3.63** |

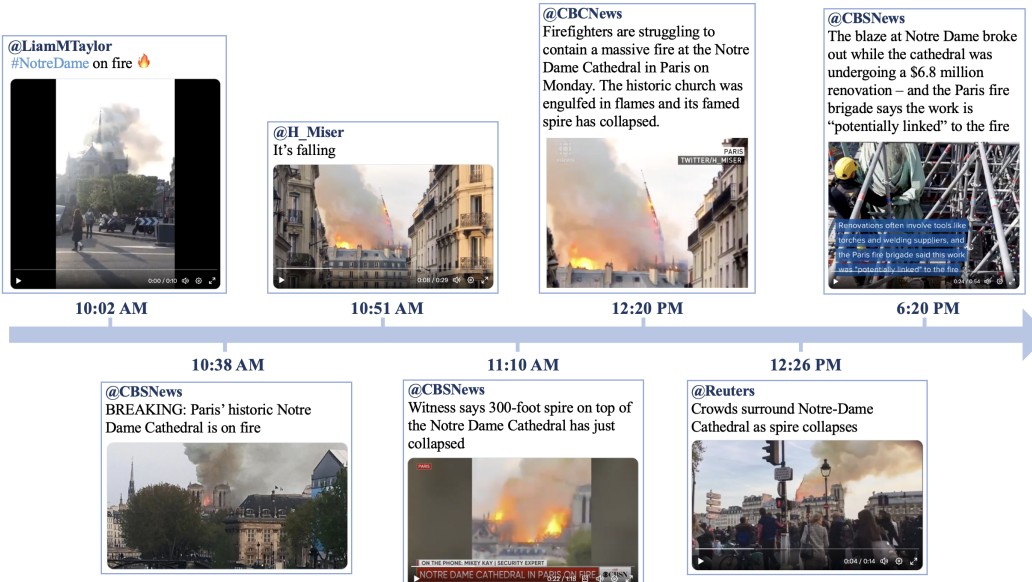

Figure 3: Snapshot of video news coverage of the 2019 Notre Dame fire news cycle (an event in MultiVENT). The fire and the fallen spire were initially reported through first-person social media video uploads (at 10:02 AM and 10:51 AM, respectively) and then later broadcast by news organizations in more detail (10:38 AM, 11:10 AM, 12:20 PM, 12:26 PM). Some news coverage (12:20 PM) directly used first-person social media footage (10:51 AM). Hours later, news agencies uploaded more complete news stories with details and context (6:20 PM). This data suggests that it is important for models to learn from both first-person videos and official news coverage at various points in the news cycle to fully construct a factual model of the event, especially if the model is attempting to construct an event model while information develops online.

cycles for slowly unfolding events and for sudden, unexpected events that take time to assess. This is illustrated in Figure 3, which shows a snapshot of the 2019 Notre Dame fire news cycle and demonstrates how unedited and poorly curated footage, often first-person witness accounts on social media, can be instrumental in the construction of our collective understanding of events. So, we propose that teaching models to understand different video formats, despite clear discrepancies in the amount of information they present, is important for developing robust systems.

# 5 Experiments

## 5.1 Approach

We consider the problem of teaching a model to map multilingual, natural language queries to multilingual video clips. Specifically, we consider a video set $V$ and query set $T$ with a indicator mapping function $f$ that returns whether a query $t \in T$ describes a video $v \in V$. The model $h$ is provided with the full set of videos $V$ and a text query $t \in T$, and for each video $v \in V$ returns the probability that $t$ describes $v$, or $h(v, t) = \mathbb{P}[f(v, t) = 1]$. When there is a bijective mapping between queries and videos (e.g., when using video descriptions as queries), the model is evaluated on its recall when considering the top 1, 5, and 10 ranked videos (R@1, R@5, and R@10), as well as the median rank (MedR). When a given query may describe multiple videos, (e.g., when using event descriptions as queries), we instead evaluate the model on its precision given the top 1, 5, and 10 ranked videos (P@1, P@5, and P@10). We define these metrics as:

$$\text{Given} \quad S := \underset{V' \subseteq V : |V'| = k}{\arg\max} \sum_{v \in V'} h(v, t),$$

$$\text{R@}k = \frac{|\{s \in S \ : \ f(s, t) = 1\}|}{|\{v \in V \ : \ f(v, t) = 1\}|} \quad \text{and} \quad \text{P@}k = \frac{|\{s \in S \ : \ f(s, t) = 1\}|}{k}.$$

## 5.2 Model architecture and training

We introduce MultiCLIP, a multilingual baseline for video retrieval on MultiVENT. This model adopts a CLIP architecture with a multilingual text encoder and is trained on multilingual text-video pairs. In more detail:

We base our architecture on the pretrained LAION CLIP ViT-H/14 frozen XLM-Roberta-Large model [11], which jointly trains an image and text encoder on text-image data to learn to pair images with their captions. At test time, it produces a zero-shot linear layer based on the test input's visual features through which natural language captions can be passed in. The model architecture contains a vision encoder based on a ViT architecture [14] and a text encoder based on the the multilingual XLM Roberta large model [12]. A full overview of the CLIP architecture and pretraining can be found in the original paper [34].

In experiments using MultiCLIP, we first tokenize text descriptions using the XLM-Roberta-Large tokenizer, containing a vocabulary of over 250,000 words, and pass the tokens into MultiCLIP which produces a text embedding of size 1024. Next, we uniformly sample videos at a rate of 12 frames per video with an input size of 224x224, which the model uses to create a frame embedding of size 1024. To incorporate multilinguality into the model's frame-level features, we use a ViT architecture trained with a contrastive objective over multilingual image-caption pairs from the LAION-5B dataset [44], which is constructed from the Common Crawl archive using images and their alt-text to produce a multilingual image-text dataset with over 100 languages. We mean pool the frame embeddings to produce a final video embedding, and use the text and video features to compute a similarity matrix of videos and descriptions.

## 5.3 Retrieval baselines

We first evaluate MultiCLIP on the existing video retrieval task MSR-VTT [53] using the recall metrics described in Sec. 5.1 alongside contemporary video retrieval approaches (FrozenInTime [4], Clip2Video [16], InternVideo [49], and MPLUG-2 [52]). Results on MSR-VTT's validation set are reported in Table 4. The results indicate MultiCLIP performs well on standard video retrieval tasks, matching performance of separate text/vision pipeline models released within the last two years. It performs better than existing models that use separate text and vision pipelines (FrozenInTime [4] and Clip2Video [16]), but not as well as models that use larger architectures involving multimodal encodings (InternVideo [49] and MPLUG-2 [52]).

Table 4: MultiCLIP evaluated alongside existing video retrieval approaches on the video retrieval benchmark MSR-VTT. Results indicate that MultiCLIP performs adequately on existing retrieval tasks, achieving comparable results to existing models. It does not perform as well as architectures that use multimodal transformers for joint encodings such as InternVideo and MPLUG-2.

| Method | Year | Rank@1 | Rank@5 | Rank@10 |
|---|---|---|---|---|
| FrozenInTime [4] | 2021 | 32.5 | 61.5 | 71.2 |
| Clip2Video [16] | 2021 | 29.8 | 55.5 | 66.2 |
| InternVideo [49] | 2022 | **55.2** | **79.6** | **87.5** |
| MPLUG-2 [52] | 2023 | 53.1 | 77.6 | 84.7 |
| **MultiCLIP** | 2023 | 38.4 | 70.1 | 82.7 |

### 5.4 MultiVENT retrieval

We now evaluate MultiCLIP and related retrieval approaches on MultiVENT. We first use multilingual video descriptions as queries, and then we use English event summaries taken from the grounding text documents, meaning that one text query maps to up to ten videos. The event queries are selected by taking one to two sentences from each English event text document that describes the event most holistically. We exclusively use English queries for this section, as our annotators fluent in the other languages were not available for this task. In addition to MultiCLIP, we consider a set of contemporary video retrieval models with lightweight architectures (FrozenInTime [4], CLIP2Video [16], InternVideo [49], MPLUG [25], and a pooled CLIP model using the same setup as MultiCLIP without the additional multilingual pretraining). We argue that lightweight architectures are most appropriate for evaluating a full, pairwise set of similarity scores between text and video data of large multimodal corpora. Results are reported, partitioned on language, in Table 5.

We report the standard recall @ rank $k$ metric for retrieval on individual video queries, and precision @ rank $k$ for retrieval on event description queries. The results suggest that some existing video retrieval models may particularly struggle on this task, regardless of language. We hypothesize that this is due to a combination of the videos' length, complex semantic content, ambiguity, and frequent OCR content, as well as the long and often noisy video description queries.

While MultiVENT as a whole poses challenges to existing models, it is also clear that multilingual data may significantly impact performance on models trained primarily on English content - all models suffer a performance loss when evaluated on multilingual content (even when using English queries, as shown by the event description query results). While MultiCLIP suffers a performance loss on this data as well, comparing the standard pooled CLIP model against MultiCLIP shows that training on multilingual data does mitigate this multilingual performance loss: The two models perform comparably on English data, but MultiCLIP performs better on the multilingual content, especially when multilingual queries are used.

## 6   Limitations and Ethical Considerations

**Data quality and demographic representation**   The events selected from Google Trends and Wikipedia were identified due to their popularity among speakers of the target language, but were identified by American English-speakers. Therefore, the domain of news items covered by MultiVENT may be slightly biased towards English-speaking audiences. Additionally, our dataset is limited to videos uploaded to primarily English-speaking video hosting forums, which limits its breadth by (1) excluding videos taken in countries with strict internet censorship laws and (2) skewing the video set towards those taken by people who choose English-centric websites over alternatives such as Bilibili or RuTube. We choose these websites because they are used globally and allow for multilingual data collection from a small set of sources. Then, by retrieving both keywords from Wikipedia and Google Trends for video retrieval, we aim to consider both official descriptions of current events as well as more colloquial terms across languages. However, this limits the videos retrieved to those associated with these terms, meaning that we omit relevant videos listed under different keywords. Finally, while we check the description, date of upload, and visual contents of videos included in the dataset, it is possible that human error led to a small number of mislabeled or irrelevant videos being included in the dataset.

Table 5: Results showing the retrieval performance of video retrieval methods alongside MultiCLIP on MultiVENT. We use video descriptions and event descriptions as queries and partition results based on language. As shown, MultiVENT can be a difficult retrieval benchmark for video retrieval models even when considering only English, but the benefit of training on multilingual data is apparent when comparing MultiCLIP against the regular pooled CLIP model on non-English data.

| Method | Video description | | | | Event description | | |
|---|---|---|---|---|---|---|---|
| | R@1 | R@5 | R@10 | MedR | P@1 | P@5 | P@10 |
| English | | | | | | | |
| FrozenInTime [4] | 6.5 | 20.0 | 28.4 | 53.0 | 42.3 | 34.6 | 26.9 |
| MPLUG [25] | 4.6 | 13.9 | 18.3 | 184.5 | 32.7 | 32.7 | 33.7 |
| CLIP2Video [16] | 41.3 | 71.8 | 80.4 | 2.0 | 96.2 | **96.9** | 73.3 |
| InternVideo [49] | 53.8 | 83.1 | 88.7 | **1.0** | 94.2 | 93.5 | 79.6 |
| CLIP (pooled) [34] | **55.9** | 83.9 | 91.3 | **1.0** | 98.1 | 94.6 | **80.6** |
| **MultiCLIP** | **55.9** | **84.5** | **92.3** | **1.0** | **100.0** | **96.9** | **80.6** |
| Arabic + Chinese + Korean + Russian | | | | | | | |
| FrozenInTime [4] | 0.5 | 1.2 | 2.5 | 793.5 | 29.8 | 22.6 | 17.6 |
| MPLUG [25] | 4.4 | 12.2 | 16.4 | 315.0 | 10.6 | 10.4 | 10.5 |
| CLIP2Video [16] | 2.4 | 7.2 | 10.5 | 166.5 | 14.4 | 9.4 | 7.3 |
| InternVideo [49] | 5.7 | 13.9 | 19.8 | 91.0 | 79.3 | 71.0 | 55.7 |
| CLIP (pooled) [34] | 6.2 | 15.9 | 22.4 | 79.5 | 83.7 | 73.3 | 58.2 |
| **MultiCLIP** | **32.6** | **64.7** | **79.5** | **3.0** | **85.6** | **76.4** | **61.5** |

**Privacy concerns**   The dataset consists of videos publicly uploaded to social media websites, but the videos are not directly available via the dataset download. We provide links to the videos which may be downloaded directly from their sources, and so these videos will automatically removed from the dataset if the video owners choose to revoke public visibility access. This may affect experiment replication if a video is removed from the dataset, but a small number of removed data points is unlikely to significantly impact experiment results.

**Model misuse**   It is possible that models fine-tuned on MultiVENT without any safety or fairness methods considered may result in biased outputs or other fairness concerns. Therefore, we strongly urge researchers using this dataset as training data to employ ethical practices when training their models with this data.

**Multilingulaity and extensions**   Due to resource constraints, we focused on five diverse and widely-spoken languages that would have distinct spaces of current event coverage. However, including more languages would improve coverage, mitigate bias, and motivate more robust multilingual models. Using the methods described in Section 3, in the future the MultiVENT dataset can feasibly be scaled up to incorporate additional languages, events, and videos.

## 7   Conclusions and Future Work

We introduce MultiVENT, a multimodal, multilingual dataset grounded in natural language documents for event-centric video retrieval and information acquisition consisting of 2,396 videos covering 260 current events in five target languages paired with multilingual text documents. We use this dataset to characterize online news coverage and how models can use this online content for information acquisition. We propose a multilingual video retrieval benchmark using MultiVENT and present MultiCLIP, multilingual video retrieval model to serve as a baseline for the task. We evaluate this model and related retrieval approaches on MSR-VTT and MultiVENT to illustrate the importance of pretraining on multilingual data for evaluation on MultiVENT. In future work, we aim to explore the effect that joint vision-OCR embeddings can have on video retrieval in text-heavy contexts. Additional tasks that could be explored for MultiVENT include cross-modal entity alignment, multilingual report generation, and multilingual video captioning. Also in future work, a PAQ-adjacent system [24] for automatically extracting question-answer pairs from video content and video-document pairs could be developed and applied to MultiVENT. Through this, a framework for teaching models to perform open-domain question-answering tasks with multimodal background corpora could be established, expanding the domain of questions a model can answer.

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

| Disaster | Social | Political | Technology/Science/Business |
|---|---|---|---|
| **Arabic** | | | |
| 2020 Beirut Explosion | 2022 Dubai World Cup | 2023 Sudan Sanctions | 2018 Hashish Sarcophagus Discovery |
| 2022 Lusail Fire | 2023 World Art Dubai | 2020 Egypt Protests | 2023 DIFC Metaverse Platform |
| 2023 Saudi Arabia Floods | 2022 GITEX | 2016 Turkey Coup D'Etat Attempt | 2018 Dubai Police Bike Demo |
| 2022 Violence In Kabul | 2020 Dubai EXPO | 2021 Israel-Palestine Crisis | 2023 Saqqara Tomb Discovery |
| 2022 Cyprus Earthquake | 2018 Ramadan | 2021 Iraqi Political Crisis | 2023 LEAP 23 Conference |
| 2022 Iraq Dust Storm | 2021 Saudi National Day | 2017 Maan Al-Sanea Arrest | 2017 NEOM Project Announcement |
| 2019 Ataba Fire | 2023 Dubai Marathon | 2023 SA-Iran Relations Restoration | 2022 Dubai Metaverse Forum |
| 2016 EgyptAir Plane Crash | 2021 Riyadh Season | 2022 Pakistani PM Removal | 2023 SpaceX CCP Mission |
| 2021 Sohag Train Collision | 2021 AFCON | 2022-23 Mahsa Amini Protests | 2019 FIRST Global Challenge |
| 2021 Cairo Factory Fire | 2019 Cairo Film Festival | 2018-19 Gaza Border Protests | 2023 Saudi Arabia MS Datacenter |
| 2021 Cyclone Shasheen | 2022 Sheikh Zayed Heritage Festival | 2023 Iran-Iraq Security Deal | 2022 Saudi Arabia Global AI Summit |
| 2023 Jeddah Floods | 2019 Arabian Gulf Cup | 2016 King Abdullah Death | 2022 Uber Data Breach |
| 2020 Riyadh Drone Attack | 2022 FIFA World Cup | 2019 Ben Gurion Airport Protest | 2023 Kuwait Satellite |
| **Chinese** | | | |
| 2021 Three Gorges Dam Flood | 2022 Olympics | 2022 Hong Kong Election | 2019 Chang'E 4 Landing |
| 2018 Hualien Earthquake | 2021 "Salmon Chaos" | 2022 A4 Revolution | 2021 Baby Yingliang Discovery |
| 2021 Qixia Gold Mine Accident | 2022 China Air Show | 2022 Foxconn Factory Protests | 2019 Bytedance Lawsuit |
| 2014 Typhoon Rammasun | 2021 Taipei Rose Festival | 2019 Hong Kong Extradition | 2020 Chang'E 5 Landing |
| 2022 Qinghai Floods | 2021 Universal Studios Beijing Opening | 2023 Spy Ballon | 2023 Huawei 5.5G Announcement |
| 2022 Luding Earthquake | 2023 Taiwan Lantern Fest | 2023 Healthcare Reform Protests | 2022 Google Bard Announcement |
| 2019 Typhoon Lekima | 2022 Macau Grand Prix | 2020 Taiwan Election | 2021 Personal Information Law |
| 2020 Anshun Bus Crash | 2018 Shenzhen Open | 2023 Li Qiang Elected | 2018 Jack Ma Alibaba Abdication |
| 2015 Tianjin Explosion | 2021 Lunar New Year | 2022 Sitong Bridge Protest | 2020 Semiconductor Shortage |
| 2022 Shanghai Jinshan Fire | 2021 Hong Kong ACG | 2019 EVA Air Strike | 2020 Taiwanese Self-Driving Bus |
| 2021 Shiyan Pipeline Explosion | 2022 Qingming Festival | 2020 China-India Skirmish | 2022 Mengtian Module Launch |
| 2022 Changsha Fire | 2021 Comic World Taiwan | 2023 Missing Taiwanese Soldier | 2020-2021 EAST Developments |
| 2022 Taitung Earthquakes | 2022 Beijing International Marathon | 2018 China-US Trade War | 2021 Hualong One Reactor |
| **English** | | | |
| 2018 Lower Puna Eruption | 2022 World Series | 2022 Starbucks Strike | 2021 mRNA Vaccine Rollout |
| 2021 Oregon Fires | 2018 Royal Wedding | 2017 Catalan Protests | 2020 CRISPR Nobel Prize |
| 2019 Notre Dame Fire | 2022 San Diego Comic-Con | 2020 Brexit | 2020 Oregon Psilocybin Legalization |
| 2022 Keystone Pipeline Oil Leak | 2019 "Storm Area 51" | 2022 UK Government Crisis | 2019 Messier 87* Images |
| 2020 California Fires | 2017 Fyre Fest | 2022 Georgia Senate Race | 2022 Southwest Cancellation Crisis |
| 2018 Anchorage Earthquake | 2019 Cricket World Cup | 2019 US Government Shutdown | 2021 Inspiration4 Launch |
| 2019-20 Australia Fires | 2018 Coachella | 2021 Canadian Election | 2020 AlphaFold2 |
| 2019 Townsville Flood | 2023 Superbowl | 2022 Novak Djokovic Visa Controversy | 2022 San Francisco Waymo Trials |
| 2022 Las Vegas Floods | 2019 NBA Finals | 2023 Black Sea Drone Incident | 2020 NASA Mars Mission |
| 2017 Hurricane Irma | 2021 Oscars | 2020 US Election | 2021 James Webb Telescope Launch |
| 2022 Ferndale Earthquake | 2016 Olympics | 2022 Queen Elizabeth's Funeral | 2016 Pokemon Go Launch |
| 2020 Nashville Tornado | 2022 Wimbledon Championships | 2022 Canada Convoy Protest | 2017 Snapchat IPO |
| 2018 Sulawesi Tsunami | 2019 Met Gala | 2019-20 Trump Impeachment | 2018 Zuckerberg Senate Hearing |
| **Korean** | | | |
| 2016 East Asia Cold Wave | 2019 SEMICON | 2016-17 South Korean Protests | 2022 Nuri Rocket Launch |
| 2017 Pohang Earthquake | 2022 Ultra Korea | 2016 Choi Soon-Sil Hearing | 2023 Danuri Photos |
| 2019 Typhoon Lingling | 2022 COMEUP | 2016 GSOMIA Pact | 2022 Tesla Optimus Demonstration |
| 2018 Northeast Asia Heat Wave | 2022 Korea Open Series | 2018 Opinion Rigging Scandal | 2023 Consumer Electronics Show |
| 2020 Korean Floods | 2022 Taekwondo Championship | 2023 Japan-South Korea Trade Dispute | 2022 Hwasong-17 ICBM Launch |
| 2018 Typhoon Soulik | 2023 Jeju Fire Festival | 2022 BTS Military Draft | 2021 Anti-Google Law |
| 2022 Typhoon Nanmadol | 2022 Emmy Awards | 2018-19 President Resignation Movement | 2023 LG U+ Security Breach |
| 2022 South Korean Floods | 2022 Kimchi Festival | 2022 South Korea Presidential Election | 2014 Sewol Ferry Salvage |
| 2022 Daegu Office Fire | 2023 Arario Gallery Reopening | 2017 US TPP Withdrawal | 2017 Lotte World Tower Opening |
| 2020 Icheon Fire | 2022 Frieze Seoul Fair | 2019 DMZ Summit | 2022 Hyundai Mobed Demo |
| 2022 Typhoon Hinnamnor | 2019 FIFA Women's Cup | 2019 South Korean Capitol Attack | 2022 Korea Blockchain Week |
| 2021 East Asia Sandstorm | 2013 Asian Rowing Championship | 2016 Geun-Hye Impeachment | 2022 Tech Layoffs |
| 2016 Gyeongju Earthquake | 2016 Speed Skating Championships | 2017 South Korea Election | 2016 AlphaGO Match |
| **Russian** | | | |
| 2017 Saratov Plane Crash | 2019 Russia Comic-Con | 2022 ZNPP Capture | 2022 Communications Exhibition |
| 2022 Kostroma Cafe Fire | 2022 Grushinsky Festival | 2022 Anti-Mobilization Protests | 2021 Paxlovid Licensing |
| 2022 Soyuz MS-22 Leak | 2017 Moscow Film Festival | 2022 Ukraine Mobilization | 2022 Twitter Acquisition |
| 2017 Moscow Storm | 2019 Winter Universiade | 2014 Revolution Of Dignity | RS-24 Yars Missile |
| 2022 Crimean Bridge Explosion | 2019 Men's Ice Hockey Championship | 2018 US Syria Withdrawal | 2022 Digital Services Act |
| 2018 Kemerovo Fire | 2021 Eurovision | 2016 Russian Election | 2022 FTX Bankruptcy |
| 2020 Norilsk Oil Spill | 2022 Moscow Marathon | 2022 Bout-Griner Prisoner Exchange | 2020 Sputnik V Distribution |
| 2020 Typhoon Maysak | 2018 FIFA World Cup | 2022 Moldova Government Resignation | 2018 Crimean Bridge Opening |
| 2023 St Petersburg Explosion | 2022 Voice Of Nomads | 2019 Ingushetia Protests | 2019 Edcrunch Conference |
| 2021 Yalta Flood | 2016 World Chess Championship | 2022 St. Petersburg Economic Forum | 2022 Kamaz Jupiter 30 Announcement |
| 2021 Listvyazhnaya Mine Disaster | 2019 Afisha Picnic | 2020-21 Khabarovsk Krai Protests | 2023 Luna 25 |
| 2021 Norilsk Avalanche | 2016 Rostelecom Cup | 2019 Ukraine Election | 2021 International Lunar Research Station |
| 2018 Magnitogorsk Building Fire | 2017 J-Fest | 2020 Third Kyrgyz Revolution | 2023 Global Energy Crisis |

Table 6: All current events covered in MultiVENT, organized by language and event category. Each current event has up to ten videos and corresponding video descriptions, along with 1-2 full text documents.

# A Current events

In Table 6 we list the full set of current events covered in MultiVENT.

| Source | Arabic | Chinese | English | Korean | Russian |
|---|---|---|---|---|---|
| Mean tokens | 123 | 114 | 95 | 108 | 170 |
| Median tokens | 25 | 53 | 38 | 53 | 54 |
| Mean unique tokens | 82 | 74 | 55 | 78 | 97 |
| Total tokens | 62820 | 60863 | 48072 | 56329 | 87493 |
| Total unique tokens | 17713 | 13824 | 6473 | 20820 | 19234 |
| Num. videos w/o dialogue | 9 | 13 | 15 | 20 | 0 |

Table 7: Audio information taken from Whisper transcripts of the MultiVENT video dataset. We partition statistics on language. As shown in the table, the dataset is rich in audio content and only approximately 2% of the dataset did not have detected spoken audio content.

| FrameNet Template | Video Template |
|---|---|
| Disaster | Disaster |
| Duration | _ |
| Place | Place |
| Responder | First responders |
| Response | _ |
| Time | Time |
| Victim | Affected people |
| _ | Disaster outcomes |

Table 8: Original FrameNet disaster template compared against our revised template for video content.

# B  Audio content in MultiVENT

To assess the audio content present in the dataset, we pass the video set through OpenAI's Whisper model[35] to produce multilingual audio transcripts. We parse this data and present general statistics in Table 7. We use NLTK to tokenize all the transcripts except Chinese data which is tokenized with Jieba[1].

# C  Disaster template modification

We modify the FrameNet "disaster scenario" template, which we use as a guide to annotate the videos in our Section 4 analysis. Specifically, we remove the "duration" and "response" fields from the FrameNet template, as duration is generally not depicted in video footage and the "first responders" field typically aligns with "response" content in video. We then add a "disaster outcomes" field as video content typically includes visually relevant depictions of the disaster aftermath. The difference between the two templates is shown in Table 8.

---

[1]https://github.com/fxsjy/jieba

