# OpenReview forum: "MultiVENT: Multilingual Videos of Events and Aligned Natural Text"
_NeurIPS.cc/2023/Track/Datasets_and_Benchmarks — NeurIPS 2023 Datasets and Benchmarks Poster_

### Official Review · Reviewer_cNPz · 2023-07-10
**Multilingual Videos of Events with Aligned Natural Text**

**Rating:** 5
**Confidence:** 2
**Correctness:** The claims presented in this work app…
**Clarity:** This work is easy to comprehend and f…

**Strengths:**

Existing news video datasets primarily focus on content produced in English. However, the authors of this work have made a valuable contribution by providing a multilingual dataset consisting of videos capturing events and aligned natural text.

**Additional Feedback:**

The paper could benefit from a clearer explanation of the novelty and unique contributions of the proposed dataset compared to existing datasets in the field.

**Documentation:**

The paper would benefit from a more detailed description.

**Ethics:**

 It would be beneficial to discuss ethical considerations.

**Limitations:**

The research motivation for this dataset needs to be further clarified, and the unique aspects of this dataset should be explained in more detail.

**Opportunities For Improvement:**

It would be beneficial if the authors could also provide audio information.

**Relation To Prior Work:**

Additional discussion regarding the significance and contributions is desired in this work.

**Summary And Contributions:**

The authors have introduced a dataset consisting of multilingual, event-centric videos that are grounded in text documents across five target languages.

---

> ### Author Response · Authors · 2023-08-21
> **Thank you for your helpful review**
>
> Thank you for your helpful comments. We have updated the paper in response to reviewer feedback and address common concerns in the general comment above. We hope to clarify your specific comments and note where relevant paper updates have been made below:
>
> **It would be beneficial if the authors could also provide audio information.**
>
> We agree that this information would be beneficial to readers. We have run OpenAI’s Whisper model on the dataset and include audio statistics in the appendix of the revised paper draft. The dataset contains a mean 122 audio tokens per video and 77 unique audio tokens per video. Only approximately 2% of the videos in the dataset do not include detected spoken audio content.
>
> **The research motivation for this dataset needs to be further clarified, and the unique aspects of this dataset should be explained in more detail… Additional discussion regarding the significance and contributions is desired in this work.**
>
> Thank you for this feedback. In summary, we address the lack of multilingual news video datasets that reflect the reality of our modern information landscape by proposing a dataset of diverse web videos depicting current events across five languages, since existing news video datasets are overwhelmingly English and focused on professionally developed content.
>
> We introduce these ideas as follows: In the introduction, we describe the nature of existing news video datasets and how they fail to reflect the range of content that humans learn about current events from. We explore how underrepresented news video formats shape modern news dissemination in this section as well as in our analysis in Section 4, motivating the need for models that can interpret such data. We then explain in the introduction that we aim to address the lack of data reflecting these underrepresented formats by constructing a dataset of diverse, multilingual news videos, which we show is a difficult benchmark for video retrieval for existing models.
>
> To improve the clarity of these points in the manuscript, we have added a sentence directly before our list of contributions in the introduction summarizing the motivation and unique contributions.
>
> **The paper would benefit from a more detailed description.**
>
> In addition to the description of the dataset construction process and dataset contents statistics in Section 3, we have included the dataset’s datasheet (originally provided in the supplementary materials) in the GitHub repository linked from the main paper. We have also provided additional event information in the Appendix.
>
> **It would be beneficial to discuss ethical considerations.**
>
> Thank you for raising this concern. We have added a comprehensive “Limitations and Ethical Considerations” section to address data quality and demographic representation, privacy concerns, model misuse, multilinguality, and extensions.

---

> > ### Author Response · Authors · 2023-08-24
> >
> > Thank you again for your review. As the rebuttal period ends on Tuesday, please let us know if you have any remaining follow-up questions or feedback. Thanks!

---

### Official Review · Reviewer_D4Di · 2023-07-22
**A new multilingual, multimodal dataset for video retrieval**

**Rating:** 6
**Confidence:** 3

**Strengths:**

1.	This is the first large-scale multilingual video retrieval dataset spanning 5 languages (Arabic, Chinese, English, Korean, Russian) and containing diverse video types beyond just news broadcasts.

2.	The authors adopt CLIP to develop a simple yet effective method for multilingual, event-centric video retrieval.

3.	Methods and experiments are clearly presented, making them easy to follow.


**Additional Feedback:**

- The videos of MultiVENT focus specifically on news events of four categories (disaster events, political events, social events, and technology events). Expanding to other video domains and events could improve generalizability.



**Clarity:**

The related works are not well reviewed. A good literature review should give the whole picture of the field and the craft view of existing methods by mentioning their advantages and disadvantages or provide a discriminating appraisal of the work that has been carried out, and point out the scientific research issues behind the task itself, research flow and future insights.

**Correctness:**

The dataset is constructed in a sound way, collecting multilingual data directly rather than relying on translation. This avoids issues like translation errors and better reflects diverse online news.

**Documentation:**

The detail on data collection and organization is sufficient.No, there are no or only very minor ethics concerns

**Ethics:**

No, there are no ethics concerns.

**Limitations:**

1.	MultiVENT covers 5 languages, but a larger diversity of languages could reveal more insights into multilingual modeling. For instance, other languages, like Japanese and Vietnamese, also have a large number of speakers.

2.	The source of the data is too simple; most of it comes from YouTube and Twitter.

**Opportunities For Improvement:**

1.	The authors can develop more tasks using the proposed dataset.

2.	Compare more methods on MultiVENT. For instance, there is a lack of experiments of MPLUG-2 on MultiVENT.

3.	The author can utilize OCR to extract text within the video to help with the retrieval.

4.	Add some details on how to modify the "disaster scenario" template.

5.	The difference between MultiCLIP and CLIP is unclear. Does MultiCLIP simply change the text encoder compared to CLIP?"


**Relation To Prior Work:**

The motivation is clear. The authors clearly discuss how this work differs from previous contributions.

**Summary And Contributions:**

The paper presents MultiVENT, a new multilingual, multimodal dataset for video retrieval. The dataset contains 2,396 videos covering 260 current events in 5 languages (Arabic, Chinese, English, Korean, Russian). Additionally, the authors develop a simple yet effective method to address the problem, utilizing CLIP as a baseline for video retrieval on this dataset.

---

> ### Author Response · Authors · 2023-08-21
> **Thank you for your thorough review (1/2)**
>
> Thank you for your thorough comments and suggestions. We have updated the paper in response to reviewer feedback and address common concerns in the general comment above. We hope to clarify your specific comments and note where relevant paper updates have been made below:
>
> **Add some details on how to modify the “disaster scenario” template.**
>
> Thank you for raising this point - we agree that this modification is interesting and the details should be included in the paper. We remove the "duration" and "response" fields from the FrameNet template, as duration is generally not depicted in video footage and the "first responders" field typically aligns with "response" content in video. We then add a "disaster outcomes" field as video content typically includes visually relevant depictions of the disaster aftermath. We include this description and a table showing the differences between the two templates in the appendix of the revised paper.
>
> **The author can utilize OCR to extract text within the video to help with the retrieval.**
>
> We agree that it is highly likely that adding explicit OCR modeling to the retrieval pipeline would improve retrieval scores, as indicated by prior work in English OCR-centric video retrieval [1]. We mention the possibility of exploring this avenue more comprehensively in the Conclusions section, but leave it for a future paper as extending a method similar to the one detailed in [1] to the multilingual domain would involve substantial additional work (multilingual training data collection, model training, etc.) beyond what we aim to propose in this paper.
>
> **The authors can develop more tasks using the proposed dataset.**
>
> Thank you for this feedback. We have added a sentence to our Conclusions and Future Work section highlighting some of these potential tasks, including cross-modal entity alignment, multilingual report generation, and multilingual video captioning.
>
> **The difference between MultiCLIP and CLIP is unclear. Does MultiCLIP simply change the text encoder compared to CLIP?**
>
> MultiCLIP and CLIP are a bit more distinct: In addition to using the XLM-Roberta-Large tokenizer, we use a ViT architecture trained with a contrastive objective over multilingual image-caption pairs from the LAION-5B dataset to incorporate multilinguality into the model’s frame-level features. To adapt to the video domain, the model samples twelve frames per video and mean pools the frame embeddings to produce a final video embedding, which is then used to compute the output similarity matrix. We have emphasized this distinction in the first sentence of Section 5.2 in our revised submission.
>
> **Compare more methods on MultiVENT. For instance, there is a lack of experiments of MPLUG-2 on MultiVENT.**
>
> The authors of MPLUG-2 have not publicly released their fine-tuned model or implementation code for the downstream task of video retrieval, and so we do not include evaluations on this model. However, the authors have released this necessary information for their original MPLUG model from 2022, which we have included experiment results for in our revised Section 5.
>
> **A larger diversity of languages could reveal more insights into multilingual modeling.**
>
> We agree that more languages would improve coverage, mitigate bias, and motivate more robust multilingual models. Due to resource constraints we focused on five diverse and widely-spoken languages that would have distinct spaces of current event coverage, but even with just the existing set of current events the MultiVENT dataset could be scaled up to incorporate additional languages and videos in the future.
>
> **The source of the data is too simple; most of it comes from YouTube and Twitter.**
>
> To reduce content/demographic bias, clips in video datasets would ideally come from a balanced and diverse array of sources. We include the lack of diversity in video sources in our revised Limitations section. However, many video sources focus on videos in one or two languages, such as RuTube or BiliBili. We opted to use YouTube, YouTube Shorts, and Twitter since they are three platforms used globally that would allow us to collect multilingual content all from the same sources.

---

> > ### Author Response · Authors · 2023-08-21
> > **Thank you for your thorough review (2/2)**
> >
> > **The related works are not well reviewed.**
> >
> > In the related work section, we aim to provide a comprehensive overview of the state of the field in areas relevant to our dataset and task, namely video retrieval and report generation. Additionally, we provide additional context for the task in the introduction: For example, where we describe the range of existing news video datasets, specifically. If you could provide more detail regarding what related work (either papers or subjects) we are omitting, we would be happy to review and include them in the paper.
> >
> > **Expanding to other video domains and events could improve generalizability.**
> >
> > We agree that expanding to other video domains would improve generalizability for video retrieval models. For this paper, we specifically are interested in the topic of report generation for multimodal news content, and so we focus on the domain of news coverage. We aim to cover a wide range of event types, spanning the major categories across which news content is generated - we include a full list of these events in the revised Appendix. Within this news domain, expanding to include additional events would improve coverage within the scope of our work and would improve retrieval models fine-tuned on this data, which we leave for future work.
> >
> > [1] Wu, Weijia, Yuzhong Zhao, Zhuang Li, Jiahong Li, Hong Zhou, Mike Zheng Shou, and Xiang Bai. "A Large Cross-Modal Video Retrieval Dataset with Reading Comprehension." arXiv preprint arXiv:2305.03347 (2023).

---

> > > ### Author Response · Authors · 2023-08-24
> > >
> > > Thank you again for your review. As the rebuttal period ends on Tuesday, please let us know if you have any remaining follow-up questions or feedback. Thanks!

---

### Official Review · Reviewer_uXbS · 2023-07-22
**A new multilingual video dataset**

**Rating:** 7
**Confidence:** 5
**Clarity:** The paper is written with great clari…

**Strengths:**

1. Good literature review that is comprehensive and motivates the proposed work well.
2. Interesting insights into content evolution over time.
3. Decent baseline technique.

**Additional Feedback:**

In addition to adding arguments to justify the seemingly small number of videos in your dataset, please also show how your approach could easily scale up to more videos and languages. I feel that it can which is why I am asking you to do so.

**Correctness:**

The claims are correct. The dataset is constructed in a strong way. The evaluation methods and design are sound. My only concern is that the number of videos in the dataset is low.

**Documentation:**

The documentation is adequate and there is sufficient detail to support reproducibility.

**Ethics:**

I do not have any ethical concerns.

**Limitations:**

The authors do not seem to have addressed the limitations of their work and the potential societal impact.
Their development of multilingual data does go towards alleviating biases introduced by using only English and opens the way to larger data sets in the future that would include even more languages. Their overall technique is content neutral and is hence not affected significantly by any biases in the data. I think if the authors follow up on these and other points they will be fine. There are no serious ethical implications of their work.

**Opportunities For Improvement:**

1, While I appreciate the difficulty of video task annotation, I am just not convinced that the authors have enough videos in their dataset. I am open to being convinced otherwise.

**Relation To Prior Work:**

The literature review is thorough and shows clearly how the proposed work advances over the state of the art.

**Summary And Contributions:**

This paper describes a new dataset that is multilingual (five different languages) of grounded videos.
The contributions are:
1. A new multilingual dataset
2. New illustration of content evolution over time.
4. New baseline technique for video retrieval.

---

> ### Author Response · Authors · 2023-08-21
> **Thank you for your thoughtful comments**
>
> Thank you for your thoughtful review. We have updated the paper in response to reviewer feedback and address common concerns in the general comment above. We hope to clarify your specific comments and note where relevant paper updates have been made below:
>
> **I am just not convinced that the authors have enough videos in their dataset.**
>
> As large, pre-trained models continue to grow in popularity and “from scratch” training costs become increasingly prohibitive, high-quality, domain-specific datasets containing fewer data points are becoming increasingly beneficial in the field as researchers move towards fine-tuning existing models, like CLIP, instead of training new models from scratch. These smaller datasets are also useful for evaluating large models’ downstream performance on specific domains and tasks.
>
> This being said, we would like to highlight the amount of raw content that MultiVENT provides. MultiVENT has fewer videos than many similar retrieval benchmarks, but on average each video is much longer (mean=83.7 seconds) than those of related datasets (which often rely on short clips with length < 10 seconds). MultiVENT contains 55.7 hours of content, which is on the same scale as popular video retrieval benchmarks such as MSR-VTT (41.2 hours) and LSMDC (56 hours). Furthermore, while MultiVENT has fewer text descriptions than many datasets, these descriptions can be quite long compared to typical single sentence descriptions. In addition to the linked Wikipedia articles which contain various internal and external links, many of the video text descriptions contain external links to additional articles covering the event in more detail. These links are included in the dataset metadata and can be easily scraped for additional text content, amounting to hundreds of additional long-form text documents that can be mapped to videos.
>
> **Please also show how your approach could easily scale up to more videos and languages.**
>
> We agree that this is quite feasible given the structure of our dataset. The most time-consuming process of the dataset collection process was the curation of a balanced set of current events. Once events are selected, streamlining the process of video retrieval is not particularly time-consuming, especially if one chooses to employ AMT workers for video filtering and quality assurance.
>
> Video scaling: Given an initial search query taken from the event name or text document (both provided in the primary dataset file), a YouTube API (or other video source) can be queried for a set of $n$ videos. YouTube is convenient for this as both traditional YouTube videos (often highly-processed content) as well as YouTube Shorts (generally short, unprocessed content uploaded by non-professionals) can be retrieved. These videos can be filtered to only retrieve videos uploaded around the approximate time the event took place (retrievable metadata from the existing videos included in the dataset, and also generally listed in the paired text documents) and then manually scanned through either by a researcher or Amazon Mechanical Turk worker for quality assurance.
>
> Language scaling: By obtaining search terms for an event in $k$ additional languages via translation software and retrieving $n$ videos per query via YouTube, the dataset could easily be extended with $260\*k\*n$ videos. Scraping videos for these new terms can be done in the same manner as the video scaling process described above. No one event in our dataset has paired videos in more than one language, and so even just by using the original five MultiVENT languages and scaping ten videos per language for each event, the dataset could be scaled up with $260\*10\*4=10400$ additional videos using this procedure.
>
> **The authors do not seem to have addressed the limitations of their work and the potential societal impact.**
>
> Thank you for raising this point. We have added a comprehensive “Limitations and Ethical Considerations” section to the revised paper to address multilinguality, extensions, and data bias, as well as other facets of data quality and demographic representation.

---

> > ### Author Response · Authors · 2023-08-24
> >
> > Thank you again for your review. As the rebuttal period ends on Tuesday, please let us know if you have any remaining follow-up questions or feedback. Thanks!

---

### Official Review · Reviewer_ZCm2 · 2023-07-22
**Generally good paper, more clarification is required, the dataset documentation is not provided**

**Rating:** 6
**Confidence:** 3

**Strengths:**

The effort to create an organic multilingual dataset is appreciated and indeed important for the field.

**Additional Feedback:**

My feedback is provided in above sections.

**Clarity:**

It is well-written, but it can benefit from more clarification (see my comments above).

**Correctness:**

The authors should clarify my comments in the opportunity for improvement section.

**Documentation:**

The authors provide a url to a data page as a place holder. The authors have not provided detailed documentation.

**Ethics:**

My concern is about the local annotators and how they have been recruited. It is unclear in the paper.

**Limitations:**

No specific section on this in the paper. The authors can explain how the models using this dataset may be misused. what about the multilingual aspect of the dataset?

**Opportunities For Improvement:**

The authors should clearly define what an event mean in the context of their dataset.

The authors find passages on Wikipedia about the events in the videos, how do they evaluate and confirm the correct association of the videos and the passages is unclear. Especially that the descriptions are in five different languages! Also finding these news passages and evaluating them for the first-person smart phone recordings should be even more challenging.

It is unclear who the local annotator are. Are they paid news professionals? or just volunteers who know each language? local to where?

On page 5, the authors wrote "We exclude "where" and "when" from the set of categories that visual content should be annotated for (because identifiable depictions of "where" are present in almost every frame, and "when" in virtually none)". In news videos there could be scene changes and not all visual content showing the same scene. If that is not the case in this specific dataset, the authors should clearly describe the characteristics of the visual contents.



**Relation To Prior Work:**

It is reasonable.

**Summary And Contributions:**

The paper presents a dataset of multilingual, event-centric videos along with their text descriptions across five different languages.

---

> ### Author Response · Authors · 2023-08-21
> **Thank you for your detailed review (1/2)**
>
> Thank you for your detailed feedback. We have updated the paper in response to reviewer feedback and address common concerns in the general comment above. We hope to clarify your specific comments and note where relevant paper updates have been made below:
>
> **The authors should clearly define what an event means in the context of their dataset.**
>
> We define the term “event” in the context of our dataset as follows: We draw from Davidsonian event semantics [1] to define an event as a related set of particulars existing at a shared point in space and time. To this definition, we add the additional condition that an event must also be “newsworthy", i.e., it must have been reported as a news story by more than one news organization. We have added this definition to the preamble of Section 3.
>
> **How they evaluate and confirm the correct association of the videos and the passages is unclear… Finding these news passages and evaluating them for the first-person smart phone recordings should be even more challenging.**
>
> We have added a description of our quality assurance process in Section 3.2. We search YouTube and Twitter for videos using target keywords collected from the Google Trends search and Wikipedia, retrieving videos that are tagged with this metadata, and then check the upload date to ensure that it aligns with the time of the event’s occurrence. Additionally, we skim through the video content to confirm that it is relevant to the target event. While this last step naturally involves more uncertainty when quality-checking low-quality content such as phone recordings, the keyword presence in metadata and the matching upload time are not affected by the video quality and are typically enough to confirm video relevance alongside the video skimming. We note that this process introduces data curation bias which we note in the new Limitations section of the draft (Section 6).
>
> **It is unclear who the local annotators are… My concern is about the local annotators and how they have been recruited. It is unclear in the paper.**
>
> Thank you for raising this point. Our annotation efforts were carried out by professionals with experience in data analysis, who assist with our research as part of their professional duties. They are therefore reasonably compensated for the effort. They choose themselves which tasks to participate in, whereupon we the authors provide face to face training and feedback on their activities, with a goal of maximizing correlation of their annotations with our own judgments. Their identities are not captured in any aspect of the data analysis or data being distributed. They are made fully aware of the purpose of the data as part of their decision to participate in the task, with the ability to ask questions as needed. We have clarified these points in Section 4.1.
>
> **In news videos there could be scene changes and not all visual content showing the same scene. (Concerning the choice to exclude “where” and “when” from the set of visual roles to consider in Section 4 analysis.)**
>
> This is correct - news videos often include multiple scenes that depict different backgrounds that likely provide more information relating to the location of the event compared to other formats. However, due to the very high prevalence of location-centric content across all three video formats, it is difficult to quantify exactly how much more location information is provided by one video over another, and so we omit this statistic for simplicity. This omission is a limitation of the analysis and we have stated so in the revision of Section 4.1.

---

> > ### Author Response · Authors · 2023-08-21
> > **Thank you for your detailed review (2/2)**
> >
> > **The authors can explain how the models using this dataset may be misused… What about the multilingual aspect of the dataset?**
> >
> > Thank you for raising these important points about ethical considerations. We have added a comprehensive “Limitations and Ethical Considerations” section to the revised paper to address model misuse and multilinguality, as well as data quality, demographic representation, and privacy concerns. To answer your questions here:
> >
> > It is possible that models fine-tuned on MultiVENT without considering any safety or fairness methods may result in biased outputs or other fairness concerns. In the paper, we have added a statement discussing this and cautioning researchers to employ ethical practices when training their models with this data. For multilinguality, we focused on five diverse and widely-spoken languages that would have distinct spaces of current event coverage, but including more languages would improve coverage, mitigate bias, and motivate more robust multilingual models. We argue that MultiVENT can be scaled up to incorporate additional languages without too much trouble: The event names included in the dataset file could be translated to produce video query terms for other languages that could be used to scrape more content via the YouTube (or other website) API.
> >
> > **The authors have not provided detailed documentation.**
> >
> > In the supplementary materials, we include a datasheet following the format introduced by Gebru et al. (2021) that provides detailed documentation regarding the origins, purpose, distribution, and maintenance for the MultiVENT dataset. We have uploaded this document to the GitHub repository linked in the main paper for easy access.
> >
> >
> > [1] Maienborn, Claudia. "Event semantics." by C. Maienborn, K. von Heusinger, P. Portner.—Berlin (2011): 232-266.

---

> > > ### Author Response · Authors · 2023-08-24
> > >
> > > Thank you again for your review. As the rebuttal period ends on Tuesday, please let us know if you have any remaining follow-up questions or feedback. Thanks!

---

> > > > ### Comment · Reviewer_ZCm2 · 2023-08-29
> > > >
> > > > The authors have addressed my concern. I keep my rating.

---

### Author Response · Authors · 2023-08-21
**General Response**

We thank the reviewers for their thoughtful responses to our paper. We noticed a few common concerns that we would like to address here. We are posting individual responses to each reviewer to address their unique comments individually.


**We would like to elaborate on our data analysis done with local annotators (Reviewers wkoj and ZCm222).**

We would first like to emphasize that the human annotations collected for this paper are for Section 4 analysis only and are not included in the downloadable MultiVENT dataset.

Our annotation efforts were carried out by professionals with experience in data analysis, who assist with our research as part of their professional duties. They are therefore reasonably compensated for the effort. They choose themselves which tasks to participate in, whereupon we the authors provide face to face training and feedback on their activities, with a goal of maximizing correlation of their annotations with our own judgments. Their identities are not captured in any aspect of the data analysis or data being distributed. They are made fully aware of the purpose of the data as part of their decision to participate in the task, with the ability to ask questions as needed. We have summarized these points in Section 4.1.

**We have addressed other ethical considerations and potential dataset limitations in a “Limitations and Ethical Considerations” section to the revised manuscript (Reviewers wkoj, ZCm222, uXbS22, and cNPz10).**

We have added a thorough “Limitations and Ethical Considerations” section to the revised paper to address data quality, demographic representation, privacy concerns, model misuse, multilinguality and dataset extensions. We have addressed reviewers’ individual concerns regarding ethical considerations individually in our responses below.

**We would like to respond to concerns regarding dataset size, both in terms of # of videos as well as # of languages and events (Reviewers uXbS22 and D4Di22).**

As large, pre-trained models continue to grow in popularity and “from scratch” training costs become increasingly prohibitive, high-quality, domain-specific datasets containing fewer data points are becoming increasingly beneficial in the field as researchers move towards fine-tuning existing models, like CLIP, instead of training new models from scratch. These smaller datasets are also useful for evaluating large models’ downstream performance on specific domains and tasks.

This being said, we would like to highlight the amount of raw content that MultiVENT provides. MultiVENT has fewer videos than many similar retrieval benchmarks, but on average each video is much longer (mean=83.7 seconds) than those of related datasets (which often rely on short clips with length < 10 seconds). MultiVENT contains 55.7 hours of content, which is on the same scale as popular video retrieval benchmarks such as MSR-VTT (41.2 hours) and LSMDC (56 hours). Furthermore, while MultiVENT has fewer text descriptions than many datasets, these descriptions can be quite long compared to typical single sentence descriptions. In addition to the linked Wikipedia articles which contain various internal and external links, many of the video text descriptions contain external links to additional articles covering the event in more detail. These links are included in the dataset metadata and can be easily scraped for additional text content, amounting to hundreds of additional long-form text documents that can be mapped to videos.

Furthermore, we believe that MultiVENT provides necessary scaffolding for streamlined data collection to extend the dataset to include more videos and languages. The most time-consuming process of the dataset collection process was the curation of a balanced set of current events. Once events are selected, streamlining the process of video retrieval is not particularly time-consuming, especially if one chooses to employ AMT workers for video filtering and quality assurance. Once query terms are identified (directly from the dataset or via Google Translate, etc.), the YouTube API can be used to retrieve a set of candidate videos that can be filtered based on time, description, and by eye by researchers or AMT workers.

**Regarding dataset documentation (Reviewers ZCm222 and cNPz10):**

In the supplementary materials, we include a datasheet following the format introduced by Gebru et al. (2021) that provides detailed documentation regarding the origins, purpose, distribution, and maintenance for the MultiVENT dataset. We have uploaded this document to the GitHub repository linked in the main paper for easy access. Additionally, we have added detailed statistics regarding dataset contents in the Appendix.

---

### Author Response · Authors · 2023-08-21
**Paper Revision Notes**

We have updated the submitted paper, and list all notable modifications and additions here:

- Recent related work has been added and slightly reorganized in Section 2.
- A “Limitations and Ethical Considerations” section has been added (Section 6). This section discusses dataset biases, demographic representation, data source limitations, privacy concerns, other data quality concerns, model misuse, and multilinguality.
- We add information regarding human annotations in Section 4.1.
- We explicitly define “event” as it is used in the paper in Section 3.1.
- We describe the data quality assurance process in Section 3.2.
- We include the total number of hours of dataset footage in the Section 3 preamble.
- We mention additional tasks MultiVENT could be used for in Section 7.
- We add a sentence elaborating on the differences between CLIP and MultiCLIP in Section 5.2.
- We describe how the disaster template was modified for the data analysis (Section 4) in the Appendix.
- We add the full list of events in the Appendix.
- We add MPLUG retrieval results on MultiVENT to Section 5.3.
- We add a summarizing sentence at the end of the introduction.
- We add MultiVENT audio statistics to the Appendix.
- We flipped the direction of the bar graph (A2) in Figure 2.

---

### Decision · Program_Chairs · 2023-09-22

**Decision:**

Accept (Poster)

**Comment:**

Overall reviewers agree that the multilingual aspect and the diversity of news videos in this dataset represent valuable contributions to the field. Concerns about insufficient discussion of ethical considerations are well addresses in the new “Limitations and Ethical Considerations” section added during rebuttal. The authors are encouraged to incorporate the detailed suggestions provided by reviewers in the final version of the paper.